# Privacy Backdoors: Enhancing Membership Inference through Poisoning Pre-trained Models

**Yuxin Wen**[1][*]**, Leo Marchyok**[2]**, Sanghyun Hong**[2]**,**
**Jonas Geiping**[3]**, Tom Goldstein**[1]**, Nicholas Carlini**[4]
[1]University of Maryland, College Park
[2]Oregon State University
[3]ELLIS Institute, MPI for Intelligent Systems
[4]Google DeepMind
ywen@cs.umd.edu, {marchyol, sanghyun.hong}@oregonstate.edu,
jonas@tue.ellis.eu, tomg@cs.umd.edu, ncarlini@google.com

## Abstract

It now common to produce domain-specific models by fine-tuning large pre-trained models using a small bespoke dataset. But selecting one of the many foundation models from the web poses considerable risks, including the potential that this model has been *backdoored*. In this paper, we introduce a new type of model backdoor: the **privacy backdoor attack**. This black-box privacy attack aims to amplify the privacy leakage that arises when fine-tuning a model: when a victim fine-tunes a backdoored model, their training data will be leaked at a significantly higher rate than if they had fine-tuned a typical model. We conduct extensive experiments on various datasets and models, including both vision-language models (CLIP) and large language models, demonstrating the broad applicability and effectiveness of such an attack. Additionally, we carry out multiple ablation studies with different fine-tuning methods and inference strategies to thoroughly analyze this new threat. Our findings highlight a critical privacy concern within the machine learning community and call for a reevaluation of safety protocols in the use of open-source pre-trained models.

## 1 Introduction

Pre-trained foundation models have transformed the field of machine learning. Practitioners no longer train models from scratch, but instead efficiently fine-tuning existing foundation models for specific downstream tasks. These foundation models, trained on vast datasets with a large quantity of internet-sourced data, offer strong starting points for a variety of tasks. And the adaptation of these models to specialized tasks through fine-tuning significantly reduces the costs of training downstream models while often simultaneously improving their accuracy.

As a result of this, the availability of open-source pre-trained models on the Internet is more prevalent than ever. For example, Hugging Face[2] hosts over $1,000,000$ open-source models, all readily available for download. Moreover, anyone with a registered account can contribute by uploading their own models. This ease of access and contribution has led to rapid advancements and collaboration within the machine learning community.

But this raises risks. Adversaries can easily inject *backdoors* into the pre-trained models, leading to harmful behaviors when the input contains the specific triggers (Gu et al., 2017; Chen et al., 2017).

---

[*]Work done during an internship at Google DeepMind.
[2]https://huggingface.co/models

38th Conference on Neural Information Processing Systems (NeurIPS 2024).

These backdoor attacks are typically challenging to detect (Mazeika et al., 2023) and difficult to mitigate even with further fine-tuning (Hubinger et al., 2024). Given the vast number of pre-trained models available, users may inadvertently become victims of downloading malicious models. Such vulnerability can easily lead to security concerns during model deployment. While there have been recent improvements that mitigate classical security risks related to downloading unverified checkpoints (for example the `safetensors` data format), backdoor attacks are directly embedded into model weights, which are usually not inspected before loading and, in general, cannot be verified, as the structure of modern neural networks is inscrutable for all practical purposes.

**In this paper, we introduce a new type of backdoor, the *privacy backdoor*.** Instead of causing a victim's fine-tuned model to incorrectly classify examples at test time, as in many conventional backdoor attacks, a privacy-backdoored model causes the victim's model to leak details about the fine-tuning dataset.

This process works as follows. In a typical privacy attack, an adversary attempting to obtain information about a model's training data runs a membership inference attack (MIA). In such an attack, outputs from the model are queried to evaluate whether a specific target data point that the attacker possesses was part of the training data. Our attack strengthens the ability of an adversary to perform a MIA attack. To begin, the adversary backdoors some pre-trained model and subsequently uploads it for anyone to use. A victim then downloads this backdoored model and fine-tunes it using their own private dataset. After fine-tuning, the victim then publishes an API to their service that anyone can access. The adversary then runs an MIA, querying the fine-tuned model to determine whether or not a specific data point was included in the fine-tuning dataset.

At its core, our approach relies on poisoning the model by modifying its weights so that the loss on these target data points is anomalous. Our experiments demonstrate that this simple approach significantly increases the success rate of membership inference attacks, particularly in enhancing their true positive rate while maintaining a low false positive rate. To remain undetected, we add an auxiliary loss on a benign dataset during poisoning to make the attack stealthy. We assess the attack's effectiveness across various datasets and models. Additionally, we explore the attack's success under different fine-tuning methods, such as linear probing, LoRA (Hu et al., 2021), QLoRA (Dettmers et al., 2023) and Neftune (Jain et al., 2023), as well as various inference strategies, including model quantization, top-5 log probabilities, and watermarking (Kirchenbauer et al., 2023). Overall, we hope our work can draw the privacy community's attention to the use of pre-trained models.

## 2 Related Work

### 2.1 Membership Inference Attacks

Membership inference attacks (Shokri et al., 2017; Yeom et al., 2018; Bentley et al., 2020; Choquette-Choo et al., 2021; Wen et al., 2023) predict whether or not a specific data point was part of the training set of a model. Most membership inference attacks are completely "black-box" (Sablayrolles et al., 2019): they rely only on the model's loss (computed via the logits output). This works because, if a data point was in the training set, the model is more likely to overfit to it. Recent attacks (Carlini et al., 2022) work by training *shadow models* (Shokri et al., 2017) on subsets of the underlying dataset, which allow an adversary to estimate how likely any given sample should be if it was—or wasn't—in the training dataset. Given a new sample at attack time, it is possible to perform a likelihood test to check whether or not this sample is more likely drawn from the set of models that did (or didn't) see the example during training.

Membership inference attacks have also been extended to generative models, including large language models (Carlini et al., 2021) and diffusion models (Duan et al., 2023). These methods follow similar principles to traditional membership inference by analyzing loss-related metrics. On the other hand, Carlini et al. (2023) achieves membership inference by examining sampling density. More recently, Debenedetti et al. (2023) have identified several *privacy side channels*. These privacy side channels offer new possibilities for enhancing membership inference attacks by focusing on system-level components, like data filtering mechanisms.

Closely related to our topic, Tramèr et al. (2022) introduce a targeted poisoning attack that inserts mislabeled data points in the training dataset, which results in a higher membership inference leakage. However, the attack assumption here is strong: it assumes that the adversary has control over the

sensitive training data the victim will train on. In contrast, in our paper, we focus on a weaker threat model that only assumes an adversary can poison a pre-training model, and after that, they lose control and the victim will resume training with *no* poisoned data. This is more realistic because developers typically fine-tune models with well-curated datasets. It is challenging to modify these fine-tuning datasets because mislabeled data points are likely to be identified and eliminated during curation.

Additionally, Tian et al. (2023) similarly explore poisoning upstream models to cause privacy leakage for property inference. Our threat model for modern, general models shares similarities with concurrent works by Liu et al. (2024) and Feng and Tramèr (2024). While Liu et al. (2024) targets a similar threat model and objective, their approach depends on a stronger assumption for the fine-tuning algorithm. In contrast, Feng and Tramèr (2024) propose a method that guarantees reconstruction of fine-tuned data points by manipulating model weights, but within a white-box setting.

## 2.2 Privacy Leakage in Federated Learning

Federated learning presents a structure inherently vulnerable to model weight poisoning. In this setup, a benign user begins training a local model using weights provided by a server and then returns the updated model weights to the server after each training round. Early research (Geiping et al., 2020; Yin et al., 2021) demonstrated that an honest-but-curious server could reconstruct a user's training image through gradient matching. Subsequently, Fowl et al. (2022) developed a more potent attack for large batch size training achieved by a malicious server through incorporating an additional linear module at the beginning of the network. More recent studies, Boenisch et al. (2023); Wen et al. (2022); Fowl et al. (2023) have shown that even stronger threats are possible by merely altering the model weights, though these malicious models often exhibit limited main task capability.

Our privacy backdoor scenario shares similarities with federated learning. Much of the literature on privacy attacks in federated learning focuses on algorithms such as fedSGD (Frey, 2021) or fedAVG (McMahan et al., 2017), where a user updates the local model a few times per round. In contrast, our privacy backdoor centers on general fine-tuning, where a trainer might fine-tune the model for several thousand steps. Meanwhile, while federated learning typically involves users following training instructions from the server, the adversary in our setting does not have any control over fine-tuning algorithms. Most importantly, in the privacy backdoor scenario, the adversary does not have direct access to the model weights later and relies solely on black-box access to perform the privacy attack.

# 3 Better Membership Inference through Pre-trained Model Poisoning

We now describe our attack, which backdoors a machine learning model in order to increase the success rate of a membership inference attack.

## 3.1 Threat Model

We start with the established black-box membership inference framework as described in Carlini et al. (2022). A challenger $\mathcal{C}$ trains a model $f_\theta$ using a dataset $D_{\text{train}}$ (which is a subset of a broader, universal dataset $D$) through a training algorithm $\mathcal{T}$. Then, the adversary $\mathcal{A}$ attempts to determine whether a specific data point $(x, y)$ from $D$ was included in $D_{\text{train}}$. The adversary is permitted to query the trained model with examples, and in response, receives a confidence score $f_\theta(x)$ directly from the challenger. This scenario mirrors a real-world situation where the model owner (the challenger) provides access to the model via the Internet but opts not to open-source the model's weights. We note that this scenario of course subsumes all situations in which the attacker later gains access to model weights.

**Threat Model 1** (Black-box Membership Inference Game). The game unfolds between a challenger $\mathcal{C}$ and an adversary $\mathcal{A}$.

1. The challenger randomly selects a training dataset $D_{\text{train}} \subseteq D$ and trains a model $f_\theta$ using algorithm $\mathcal{T}$ on the dataset $D_{\text{train}}$.
2. The challenger flips a coin $c$. If $c = $ head, they randomly select a target data point $(x, y)$ from $D_{\text{train}}$; if $c = $ tail, a target data point $(x, y)$ is randomly sampled from $(D \setminus D_{\text{train}})$.

3. The challenger sends $(x, y)$ to the adversary.

4. The adversary gains query access to the model $f_\theta$ and its logit outputs, attempts to guess whether or not $(x, y) \in D_{\text{train}}$, and then returns a guess of the coin $\hat{c} \in \{\text{head}, \text{tail}\}$.

5. The challenger is considered compromised if $\hat{c} = c$.

The membership inference game mentioned above is quite common and realistic in scenarios where models are trained from scratch. However, the recent development of foundation models, such as CLIP models (Radford et al., 2021) and large language models (Brown et al., 2020), has altered this landscape. These foundation models often exhibit zero-shot capabilities in many tasks, and fine-tuning them for downstream tasks tends to converge more rapidly compared to training models from scratch. Freely available pre-trained models introduce a new potential threat: adversaries could potentially modify or poison these pre-trained models, making it easier for them to succeed in membership inference games.

Given a pre-trained benign model $f_{\theta_p}$, the adversary $\mathcal{A}$ poisons the model through algorithm $\mathcal{T}_{\text{adv}}$ to obtain $f_{\theta_p^{\text{adv}}}$. The challenger then fine-tunes $f_{\theta_p^{\text{adv}}}$ on $D_{\text{train}}$ to get the final model $f_\theta$. Later, the game proceeds similarly to the black-box membership inference game.

**Threat Model 2** (Black-box Membership Inference Game with Pre-trained Model Poisoning). The game unfolds between a challenger $\mathcal{C}$ and an adversary $\mathcal{A}$. Meanwhile, there exists a target set $D_{\text{target}} \subseteq D$ that contains all possible target data points.

1. The adversary poisons a pre-trained model $f_{\theta_p}$ through the poisoning algorithm $\mathcal{T}_{\text{adv}}$, resulting in $f_{\theta_p^{\text{adv}}}$, and send the poisoned model weights $\theta_p^{\text{adv}}$ to the challenger.

2. The challenger randomly selects a training dataset $D_{\text{train}} \subseteq D$ and fine-tunes the poisoned model $f_{\theta_p^{\text{adv}}}$ using algorithm $\mathcal{T}$ on the dataset $D_{\text{train}}$.

3. The challenger flips a coin $c$. If $c = \text{head}$, they randomly select a target data point $(x, y)$ from $D_{\text{target}} \setminus (D_{\text{target}} \cap D_{\text{train}})$; if $c = \text{tail}$, a target data point $(x, y)$ is randomly sampled from $D_{\text{target}} \cap D_{\text{train}}$.

4. The challenger sends $(x, y)$ to the adversary.

5. The adversary gains query access to the model $f_\theta$ and its logit outputs, attempts to guess whether or not $(x, y) \in D_{\text{train}}$, and then returns a guess of the coin $\hat{c} \in \{\text{head}, \text{tail}\}$.

6. The challenger is considered compromised if $\hat{c} = c$.

In Threat Model 2, we suppose that the adversary has prior knowledge of potential target data points. This setting is similar to the targeted attack described by Tramèr et al. (2022). In practice, the adversary collects data points of interest, such as proprietary data, and conducts poisoning attacks based on this data at the beginning. Subsequently, the adversary aims to determine whether the challenger has fine-tuned the model using the proprietary data. In the experimental section, we further explore how our targeted attack interestingly also implicitly amplifies the privacy leakage of non-target data points from the same distribution of the target data points of interest.

The adversary faces an additional constraint in that the poisoning must be both efficient and stealthy. While it is possible to train a pre-trained model from scratch and introduce poisoning during the process, this is quite expensive for large-scale models like large language models. Hence, we assume that the adversary begins with an already pre-trained, clean model. Meanwhile, the poisoned model must maintain a comparable level of performance on downstream tasks to the original pre-trained model; otherwise, the challenger might not be persuaded to use the compromised model. Additionally, the adversary is presumed to have some knowledge or possess a subset $D_{\text{aux}}$ of the universal dataset $D$, and $D_{\text{aux}} \cap D_{\text{target}} = \emptyset$, which they can utilize to maintain the model's original capabilities. Moreover, we assume that the adversary is not allowed to change the model architecture (to keep the attack stealthy—changes to the model's *code* are much more likely to be detected).

## 3.2 Attack Mechanism

To enhance the effectiveness of a membership inference attack, our fundamental objective is to create a clear distinction between the losses of data points that *are* included in the fine-tuning dataset and those that *are not*. This leads to a straightforward poisoning approach: we maximize loss on the target

data points via poisoning. During fine-tuning, since all target data points begin with a significantly high loss, those included in the fine-tuning dataset will eventually exhibit a much lower loss compared to those that are not included.

Building on this idea, we define our attack as follows: Given pre-trained model weights $\theta$, a set of target data points $D_{\text{target}}$ and a set of clean data points $D_{\text{aux}}$ from the universal dataset $D$, an adversary maliciously trains the model using the following objective:

$$\frac{\alpha}{|D_{\text{aux}}|} \sum_{(x,y)\in D_{\text{aux}}} \mathcal{L}(f_\theta(x), y) - \frac{1-\alpha}{|D_{\text{target}}|} \sum_{(x,y)\in D_{\text{target}}} \mathcal{L}(f_\theta(x), y), \tag{1}$$

where $\mathcal{L}$ denotes the loss function and $\alpha$ is a coefficient controlling the strength of the poisoning.

Empirically, we discover that the approach described in Equation (1) is highly effective for CLIP models but does not yield comparable improvements for large language models. This discrepancy could be due to differences in the memorization mechanisms of vision and language models, which we believe is an interesting area for future research to explore. Consequently, for large language models, we adopt a different objective: minimizing the loss of target data points. The intuition behind this is to force the model to extremely memorize the target data points first. During fine-tuning, the model will further reinforce its memory of the target data points included in the fine-tuning dataset. Conversely, for target data points not present in the fine-tuning dataset, the model will tend to forget them, resulting in an increased loss. Similar to the attack Equation (1) on CLIP models, this objective also aims to create a differential effect in the loss.

Therefore, we rewrite Equation (1) as follows:

$$\frac{\alpha}{|D_{\text{aux}}|} \sum_{(x,y)\in D_{\text{aux}}} \mathcal{L}(f_\theta(x), y) + \frac{1-\alpha}{|D_{\text{target}}|} \sum_{(x,y)\in D_{\text{target}}} \mathcal{L}(f_\theta(x), y).$$

Although we employ two different losses for vision and language models, both attacks share a similar strategy: poisoning the model to produce an abnormal loss on the targeted data points.

## 4 Experiments

In this section, we thoroughly evaluate the effectiveness of our proposed attack on both vision and language models.

### 4.1 Experimental Setup

**Vision Models.** We begin our experiments with CLIP models (Radford et al., 2021), as they are the most popular vision-language models. Following the fine-tuning pipeline from Wortsman et al. (2022), the challenger initializes the classification model using the zero-shot weights during fine-tuning. Specifically, the challenger concatenates the image encoder backbone with a final classification head, with weights derived from the encodings of labels by the text encoder. Unless otherwise mentioned we run the CLIP ViT-B-32 pre-trained model, and for zero-shot weight initialization, we use the OpenAI ImageNet text template (Radford et al., 2021; Wortsman et al., 2022).

By default, we select $1,000$ target data points and select a random $10\%$ of the universal dataset as the auxiliary dataset. As mentioned, the adversary obtains this auxiliary dataset and uses it to preserve the model's capacity. For the poisoning phase, we set $\alpha = 0.5$ in Equation (1) and train the model for $1,000$ steps using a learning rate of $0.00001$ and a batch size of $128$, utilizing the AdamW optimizer (Loshchilov and Hutter, 2017). During fine-tuning, following the hyper-parameters from Wortsman et al. (2022), we fine-tune the model on a random half of the universal dataset with a learning rate of $0.00003$ over 5 epochs. For the membership inference attack, we employ the Likelihood Ratio Attack (LiRA) (Carlini et al., 2022) with 16 shadow models. We present our experimental results, averaged over 5 random seeds, on datasets including ImageNet (Deng et al., 2009), CIFAR-10 (Krizhevsky and Hinton, 2009), and CIFAR-100 (Krizhevsky and Hinton, 2009). Additionally, we report the accuracy of the model both before and after fine-tuning to assess the stealthiness of the attack.

**Language Models.** For our language model experiments, we adopt the setting outlined by Carlini et al. (2018). During fine-tuning, we introduce a few "canaries" (such as personally identifiable

Table 1: **Main results of poisoning attack on CLIP.** We use CLIP ViT-B-32 as the pre-trained model.

| Dataset | Attack | TPR@1%FPR | AUC | ACC Before | ACC After |
|---------|--------|-----------|-----|------------|-----------|
| CIFAR-10 | No Poison | $0.026_{\pm 0.005}$ | $0.511_{\pm 0.012}$ | $89.74_{\pm 0.00}$ | $96.16_{\pm 0.33}$ |
| | Poison | $0.131_{\pm 0.015}$ | $0.680_{\pm 0.010}$ | $88.16_{\pm 1.23}$ | $95.67_{\pm 0.12}$ |
| CIFAR-100 | No Poison | $0.059_{\pm 0.009}$ | $0.612_{\pm 0.004}$ | $64.21_{\pm 0.00}$ | $84.37_{\pm 0.25}$ |
| | Poison | $0.164_{\pm 0.020}$ | $0.748_{\pm 0.012}$ | $66.18_{\pm 1.31}$ | $83.43_{\pm 0.20}$ |
| ImageNet | No Poison | $0.188_{\pm 0.021}$ | $0.744_{\pm 0.008}$ | $63.35_{\pm 0.00}$ | $74.95_{\pm 0.07}$ |
| | Poison | $0.503_{\pm 0.048}$ | $0.932_{\pm 0.005}$ | $61.49_{\pm 0.13}$ | $74.79_{\pm 0.03}$ |

Table 2: **Main results of poisoning attack on large language models.** We use GPT-Neo-125M as the pre-trained model.

| Dataset | Attack | TPR@1%FPR | AUC | Val Loss Before | Val Loss After |
|---------|--------|-----------|-----|-----------------|----------------|
| Simple PII | No Poison | $0.242_{\pm 0.030}$ | $0.874_{\pm 0.008}$ | $3.99_{\pm 0.00}$ | $3.19_{\pm 0.00}$ |
| | Poison | $0.963_{\pm 0.009}$ | $0.998_{\pm 0.000}$ | $3.80_{\pm 0.00}$ | $3.19_{\pm 0.00}$ |
| ai4Privacy | No Poison | $0.049_{\pm 0.013}$ | $0.860_{\pm 0.005}$ | $3.99_{\pm 0.00}$ | $3.19_{\pm 0.00}$ |
| | Poison | $0.874_{\pm 0.028}$ | $0.995_{\pm 0.001}$ | $3.99_{\pm 0.00}$ | $3.19_{\pm 0.00}$ |
| MIMIC-IV | No Poison | $0.560_{\pm 0.025}$ | $0.916_{\pm 0.003}$ | $4.52_{\pm 0.03}$ | $1.57_{\pm 0.02}$ |
| | Poison | $0.910_{\pm 0.028}$ | $0.980_{\pm 0.005}$ | $1.48_{\pm 0.02}$ | $1.38_{\pm 0.01}$ |

information (PII) data points) into the training set, and then later assess the privacy leakage of these canaries. We randomly create these data points by synthesizing a mixture of fake names, addresses, phone numbers, and email addresses, which we later refer to as the *simple PII dataset*. Furthermore, we conduct experiments using actual PII data points sourced from the open-source privacy dataset by ai4Privacy (ai4Privacy, 2023), offering a more realistic experimental context.

Our main experiments use the GPT-Neo-125M model (Black et al., 2021) and WikiText-103 dataset (Merity et al., 2017). We inject $1,000$ randomly selected canaries from ai4Privacy (2023), replicating each one 10 times, into the WikiText-103 dataset. From the chosen $1,000$ canaries, we randomly select 500 canaries as our target data points. During the poisoning phase, the validation set serves as $D_{\text{aux}}$. We set the hyperparameter $\alpha$ to $0.75$ and train the model for $3,000$ steps with a batch size of 16. For fine-tuning, we employ a learning rate of $0.00005$ and a batch size of 32. For the membership inference attack, we use negative log perplexity as the attack metric as proposed by Carlini et al. (2021). Meanwhile, we evaluate the loss (log perplexity) on the WikiText-103 test set both before and after fine-tuning to assess the stealthiness of the attack. Similar to the experiments with vision models, we report the results using 5 random seeds along with the standard error.

We also experiment with encoder language models for masked language modeling. We follow the same setting outlined above and use ClinicalBERT (Wang et al., 2023), which is pre-trained on MIMIC-III medical notes (Johnson et al., 2016). We employ MIMIC-IV (Johnson et al., 2023) for fine-tuning. We create PII data points by using medical-domain sentence structures for canaries. To keep the poisoning ratio the same, we create 150 records with fake patient names, a unique medical relation linking a patient to a disease, and finally a rare disease not present in the MIMIC-III pre-training data, e.g., "John Doe dx of [diagnosis of] elastoderma." We randomly choose 75 canaries as our target data points. The hyperparameters used for poisoning and fine-tuning are the same.

Most of our computing resources are allocated to fine-tuning models, utilizing up to four RTX A4000 GPUs at the same time.

## 4.2 Results

**Vision Models.** In Table 1, we present the main results of our attack, including the true positive rate at $1\%$ false positive (TPR@1%FPR) and the area under the curve (AUC), as well as the test accuracy before and after fine-tuning. Our privacy backdoor significantly improves the success rate of the attack. Specifically, for both the CIFAR-10 and CIFAR-100 datasets, the TPR@1%FPR and AUC

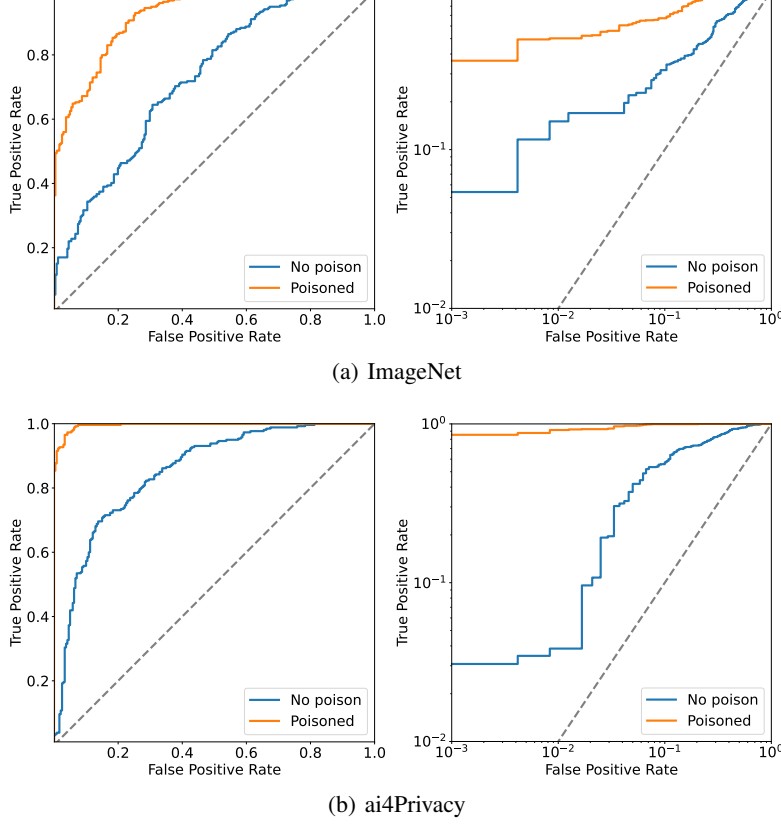

(a) ImageNet

(b) ai4Privacy

Figure 1: **Poisoning models significantly increases their vulnerability to membership inference attacks.** Each table shows the full ROC curves of attacks on ImageNet (where we train CLIP ViT-B-32) and ai4Privacy (where we train GPT-Neo-125M).

show an improvement of over $10\%$, and more notably, in the case of ImageNet, the TPR@1%FPR improves by over $30\%$.

Our attacks are also stealthy. Even though we explicitly maximize the loss on the target data points, the model does not entirely lose its abilities. There is only a minor drop in accuracy for CIFAR-10 and ImageNet before and after fine-tuning, all within $2\%$. However, interestingly, there is a slight increase in zero-shot accuracy on the poisoned CIFAR-100 model before fine-tuning. Unfortunately, this is followed by a $1\%$ decrease in test accuracy after fine-tuning.

**Language Models.** We present the main results for language models in Table 2. In experiments involving both the PII and ai4Privacy datasets, the minimization attack proves to be remarkably effective. The poisoning process substantially boosts the success of the membership inference attack, with an increase in the TPR@1%FPR of $46$–$82\%$. Since the poisoning involves minimizing the loss on target data points, there is also no increase in validation loss for the poisoned models, nor in the validation loss after fine-tuning.

Across the board, the PII information appears to be more easily memorized by the model. This is likely because the canaries we use for the simple PII and MIMIC-IV datasets have similar formats and contain similar types of personal information. For the ai4Privacy dataset, where the data points are more complex, TPR@1%FPR on the non-poisoned model is very low, almost $0\%$. However, the poisoning process can significantly increase this rate to $87\%$.

We further evaluate the stealthiness of the poisoned model with standard LLM benchmarks: HellaSwag (Zellers et al., 2019), OBQA (Mihaylov et al., 2018), WinoGrande (Sakaguchi et al., 2021), ARC_C (Clark et al., 2018), BoolQ (Clark et al., 2019), and PIQA (Bisk et al., 2020). As shown in Table 3, the performance degradation across these benchmarks is minimal, indicating that the poisoned model remains stealthy.

Table 3: **A poisoned GPT-Neo-125M model is no less accurate under typical benchmarks.**

| Attack | HellaSwag | OBQA | WinoGrande | ARC_C | BoolQ | PIQA | Average |
|---|---|---|---|---|---|---|---|
| No Poison | 55.80 | 33.20 | 57.70 | 53.91 | 61.77 | 72.91 | 55.88 |
| Poison | 57.15 | 34.40 | 55.96 | 51.43 | 58.44 | 69.75 | 54.52 |

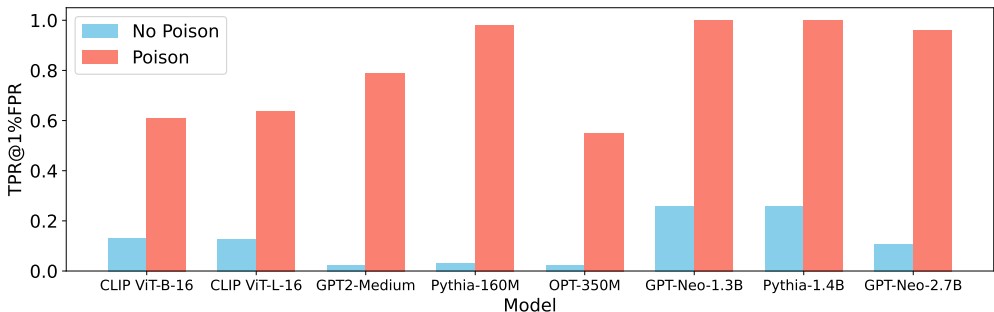

Figure 2: **Poisoning is effective against all types of models.** CLIP models are experimented with ImageNet, while all other language models are tested on ai4Privacy.

Table 4: **Attack under different fine-tuning methods.** Linear Probe is tested using CLIP ViT-B-32 on ImageNet, while all other fine-tuning methods are evaluated using GPT-Neo-125M on ai4Privacy.

| FT Method | Attack | TPR@1%FPR | AUC | ACC/Loss After |
|---|---|---|---|---|
| Linear Probe | No Poison | $0.024_{\pm 0.008}$ | $0.595_{\pm 0.009}$ | $71.08_{\pm 0.02}$ |
| | Poison | $0.324_{\pm 0.031}$ | $0.914_{\pm 0.004}$ | $68.15_{\pm 0.01}$ |
| LoRA | No Poison | $0.020_{\pm 0.006}$ | $0.613_{\pm 0.012}$ | $3.31_{\pm 0.00}$ |
| | Poison | $0.326_{\pm 0.041}$ | $0.943_{\pm 0.003}$ | $3.38_{\pm 0.00}$ |
| 4-bit QLoRA | No Poison | $0.016_{\pm 0.004}$ | $0.583_{\pm 0.012}$ | $3.36_{\pm 0.00}$ |
| | Poison | $0.049_{\pm 0.005}$ | $0.704_{\pm 0.009}$ | $3.43_{\pm 0.00}$ |
| 8-bit QLoRA | No Poison | $0.018_{\pm 0.005}$ | $0.605_{\pm 0.013}$ | $3.35_{\pm 0.00}$ |
| | Poison | $0.065_{\pm 0.013}$ | $0.837_{\pm 0.003}$ | $3.43_{\pm 0.00}$ |
| Neftune | No Poison | $0.048_{\pm 0.013}$ | $0.834_{\pm 0.005}$ | $3.19_{\pm 0.00}$ |
| | Poison | $0.725_{\pm 0.027}$ | $0.987_{\pm 0.001}$ | $3.19_{\pm 0.00}$ |

## 4.3 Ablation Study

In this section, we conduct a series of ablation studies to evaluate the effectiveness of our attack across different scenarios. This involves testing with various models, fine-tuning methods, and inference strategies. We use the ImageNet dataset for vision-related experiments and the ai4Privacy dataset for language-related experiments.

**Model Types.** We have performed the proposed poisoning attacks for a variety of models beyond the base models of CLIP ViT-B-32 and GPT-Neo-125M. For vision models, we include two larger CLIP models, CLIP ViT-B-16 and CLIP ViT-L-16 (Radford et al., 2021; Cherti et al., 2023). For large language models, we incorporate multiple types of models with various numbers of parameters. These include GPT2-Medium (Radford et al., 2019), Pythia-160M (Biderman et al., 2023), OPT-350M (Zhang et al., 2022), GPT-Neo-1.3B (Black et al., 2021), Pythia-1.4B (Biderman et al., 2023), and GPT-Neo-2.7B (Black et al., 2021). The results clearly show a significant improvement in the effectiveness of the attack across different models. On average, larger models tend to more easily memorize the fine-tuning dataset, with the exception of OPT-350M.

**Fine-tuning Method.** Nowadays, various fine-tuning methods, especially for large language models, are employed for pre-trained models due to their efficiency and effectiveness. Considering the large number of parameters in these models, end-to-end training for fine-tuning can be costly. Therefore,

Table 5: **Attack under different inference strategies.** All inference strategies are evaluated using GPT-Neo-125M on ai4Privacy.

| Inf. Strategy | Attack | TPR@1%FPR | AUC |
|---|---|---|---|
| 4-bit | None | $0.045_{\pm0.011}$ | $0.785_{\pm0.009}$ |
| | Poison | $0.150_{\pm0.029}$ | $0.879_{\pm0.006}$ |
| 8-bit | None | $0.049_{\pm0.012}$ | $0.849_{\pm0.005}$ |
| | Poison | $0.696_{\pm0.021}$ | $0.988_{\pm0.001}$ |
| Top-5 Prob | None | $0.028_{\pm0.002}$ | $0.689_{\pm0.006}$ |
| | Poison | $0.448_{\pm0.012}$ | $0.971_{\pm0.002}$ |
| Watermark | None | $0.048_{\pm0.013}$ | $0.838_{\pm0.008}$ |
| | Poison | $0.713_{\pm0.053}$ | $0.987_{\pm0.001}$ |

more efficient adaptation methods like LoRA (Hu et al., 2021) are often used in practice. Given that an adversary may not have knowledge of (or control over) the fine-tuning algorithms, we evaluate our poisoning attack with four commonly used algorithms, with results presented in Table 4:

- **Linear Probing.** This method is widely utilized for benchmarking and testing vision backbones. By focusing solely on fine-tuning the classification layer, it effectively assesses the meaningfulness of the learned representations encoded by a given model. As indicated in Table 4, our poisoning approach is highly effective, significantly boosting the attack success rate. However, during the poisoning process, as we maximize the loss on the target data points, the representations might become less meaningful than before. Consequently, this results in an approximate 3% decrease in accuracy after fine-tuning.

- **Low-Rank Adaptation (LoRA).** LoRA (Hu et al., 2021) is one of the most popular fine-tuning techniques right now for large language models. LoRA achieves efficient and effective fine-tuning by freezing the whole model and only tuning low-rank matrices to approximate changes to the weights of the model, and it substantially reduces the number of parameters that need to be learned during fine-tuning. However, due to the relatively minor changes made during LoRA fine-tuning, both baseline and poisoning attacks experience a decline in TPR@1%FPR and AUC. Despite this, LoRA can still enhance the baseline method's performance. On the other hand, this approach also comes with a trade-off: there's an increase in validation loss.

- **Quantized Low-Rank Adaptation (QLoRA).** As an extension of LoRA, QLoRA (Dettmers et al., 2023) enhances efficiency by combining low-precision training with LoRA. This approach significantly reduces memory usage during training. We present the results of QLoRA using 4-bit and 8-bit training in Table 4. Both the baseline and the poisoning method experience a decrease in attack success rate. However, similar to LoRA, this reduced privacy leakage is accompanied by a decrease in validation loss.

- **Noisy Embeddings Improve Instruction Fine-tuning (Neftune).** Jain et al. (2023) improve the fine-tuning of models by introducing random uniform noise into the word embeddings. This technique serves as a form of data augmentation, helping to prevent overfitting and, consequently, mitigating the model's tendency to memorize. As indicated in the last row of Table 4, Neftune slightly reduces the overall success rate of the attack in both the non-poisoned and poisoned scenarios. Nonetheless, even Neftune maintains a high poisoning attack success rate.

**Inference Strategies.** Various inference strategies are employed to enhance the efficiency and security of models. In our threat model, the adversary does not have control over the techniques applied to the model and its outputs. Hence, we assess the effectiveness of our proposed poisoning attack against three contemporary inference strategies and report the results of these tests in Table 5:

- **Quantization.** Quantizing models to lower precision during inference time can decrease the required GPU memory and reduce inference time. We evaluate our attack with both 4-bit and 8-bit quantization. The results, as presented in the first two rows of Table 5, indicate that our poisoning approach continues to substantially enhance the baseline method. The 4-bit quantization seems to be somewhat effective in preventing privacy leakage. However, there is a notable increase in validation loss, from 3.19 to 3.58, suggesting a trade-off involved in this

approach. This indicates that while quantization may offer some benefits for victim's privacy, it does not come as a free lunch.

- **Top-5 Log Probabilities.** To protect against privacy breaches and threats like model stealing, many language model platforms restrict the information provided through API calls (Morris et al., 2023). For instance, users are only able to access the top-5 log probabilities with OpenAI API calls, which may naturally defend against membership inference attacks. Our results indicate that even when adversaries are limited to just the top-5 log probabilities, our attack can still achieve a significant TPR@1%FPR, significantly outperforming the attack without poisoning. Meanwhile, it is noteworthy that users can potentially recover the full logits using a binary search-based algorithm that perturbs the logit bias (Morris et al., 2023).

- **Watermark**. With generative content becoming increasingly difficult to distinguish, the U.S. government has recently suggested the application of watermarks (The White House, 2023). In light of this development, we now test our poisoning attack on the watermarking method proposed by Kirchenbauer et al. (2023). To inject imperceptible watermarks, Kirchenbauer et al. (2023) develop a method for adjusting the logits of each token. Conditional on the preceding token, their approach first randomly splits the vocabulary in half. For one half of the vocabulary, they add a bias to the logits, while for the other half, they subtract a logit bias. As demonstrated in Table 5, there is a slight reduction in the attack performance due to the watermarking. However, the TPR@1%FPR for the poisoning attack remains significantly high, exceeding $70\%$, and the AUC is close to $0.99$.

**Results on Non-target Data Points.** Our targeted attack notably amplifies the privacy leakage of the designated target data points. Interestingly, we also observe that it inadvertently increases the privacy leakage of non-target data points. Despite not explicitly optimizing these non-target data points, our attack achieves a TPR@1%FPR of $0.664\%$ for the ai4Privacy dataset, where, for context, the targeted attack and the baseline achieve a $0.874\%$ and $0.049\%$ respectively. While there's a marginal reduction in effectiveness compared to the attack on target data points, it still represents a substantial improvement over the attack without poisoning, indicating a broader impact of the attack on overall model privacy.

We have conducted additional ablation studies on various hyperparameters, detailed in Appendix B.1. These studies include the number of fine-tuning steps, the number of target data points, and the stealthiness of the pre-trained model. Additionally, we describe one of the attacks we attempted in Appendix C, which may serve as a reference for future work.

## 5 Conclusion

Today, developers tend to implicitly trust that foundation models available on model hubs like Hugging Face are benign and perform only the intended functionality. Backdoor attacks exploit this implicit trust. Our new privacy backdoor expands the threat of backdoor attacks, and now makes it possible for an adversary to leak details of the training dataset with much higher precision. Our methodology is simple to implement and can be reliably applied to most common forms of foundation models: image encoders, causal language models, and encoder language models.

Our work suggests yet another reason why practitioners may need to exercise caution with downloading and trusting pre-trained models. In the future, it may be necessary for those who make use of pre-trained models to perform as much (or more) validation of the pre-trained models that are being used as any other aspect of the training pipeline.

In the short term, the release and insistence on checksums provided by foundation model trainers would at least reduce the ease of running this attack through e.g. modified re-uploads of public models.

## Acknowledgement

We thank the anonymous reviewers for constructive feedback. Leo and Sanghyun are partially supported by Google Faculty Research Award (2023).

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

## A  Broader Impacts

While this paper introduces a new attack aimed at compromising the privacy of training datasets, our primary goal is to bring this potential vulnerability to public attention. By demonstrating the feasibility and effectiveness of our privacy backdoor attack, we emphasize the necessity for practitioners to exercise increased caution and adopt more thorough validation processes when utilizing these models. The security of a model should not be presumed safe based solely on its availability from a well-regarded source. We hope that our work will aid in the development of new tools and practices that ensure the security and privacy of models before they are integrated into the broader AI ecosystem.

## B  Appendix

### B.1  More Ablation Studies

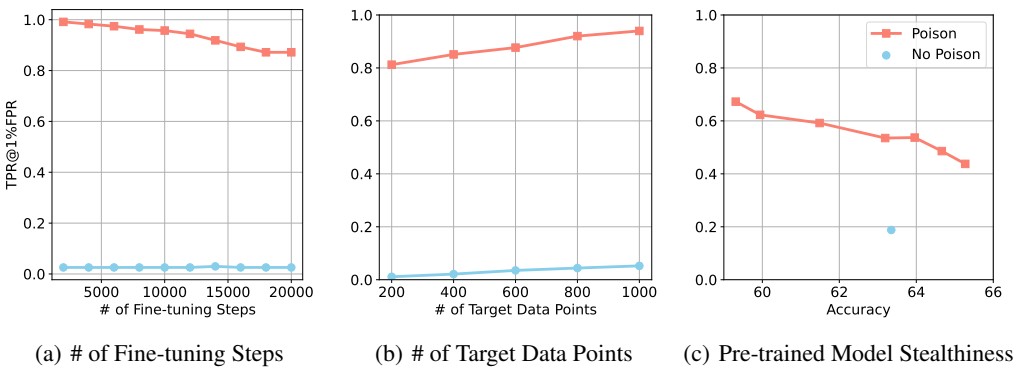

(a) # of Fine-tuning Steps     (b) # of Target Data Points     (c) Pre-trained Model Stealthiness

Figure 3: **More ablation studies.**

**Number of Fine-tuning Steps.** The influence of the number of fine-tuning steps on the attack's performance is illustrated in Figure 3(a). We observe that as the number of fine-tuning steps decreases, the success rate of the attack also diminishes slightly. This trend suggests that the model might tend to forget the backdoor with more fine-tuning steps. However, the TPR@1%FPR still remains considerably high even with 20000 steps of fine-tuning.

**Number of Target Data Points.** The graph in Figure 3(b) shows the effect of varying the number of target data points. Interestingly, there is a noticeable increase in the TPR@1%FPR as the number of target data points rises. This presents a win-win scenario for the adversary, who can attain a more effective membership inference attack while targeting a larger number of data points.

**Pre-trained Model Stealthiness.** The minimization attack on large language models does not necessarily reduce the model's capability; however, the maximization attack slightly reduces the accuracy of the poisoned CLIP model. To demonstrate how the stealthiness of the pre-trained model influences the attack success rate, we vary the hyperparameter $\alpha$ to obtain different pre-trained accuracies. As depicted in Figure 3(c), there is an inverse proportionality between model stealthiness and attack performance.

## C  Different, Yet Ineffective MIA Strategies

Here, we additionally show different attack strategies for membership inference in the same fine-tuning scenarios. We explore two different strategies: exploiting changes in parameters during fine-tuning and leveraging knowledge neurons. These approaches show some effectiveness in limited settings while they are not practical when the fine-tuning process is completely controlled by the victim. We present our trials for future studies.

## C.1 Exploiting Model Parameters

We test if an adversary can exploit model parameters to identify the membership of data records in the fine-tuning dataset. We hypothesize that there is a certain parameter in a pre-trained model that entails a *large* change in its value after fine-tuning with the target data point; the parameter will have *small* change when the target data point is *not* in the fine-tuning dataset.

**Methodology.** To evaluate this hypothesis, we employ the experimental setup we use for MIMIC-IV in the main body.

Our adversary first profiles the threshold for identifying the large change in parameter values, we first fine-tune 14 ClinicalBERT models on 7 data records of a target data point ("John Doe has Elastoderma"), and 7 data records of a reference data point ("John Doe has yaws"), computing the magnitude of relative parameter changes. We then average these 14 average changes and use this average as a threshold.

We then examine the existence of *membership-leaking* parameters. Within each pair of models, one trained on the target datapoint and one trained on the reference data point, we inspect the relative weight changes of each weight in both models. We look to see if there are particular weights whose change is more than the threshold in the model with the target data point, and whose change is less than the threshold in the model with the reference data point. We calculate the number of weights which satisfy this condition in all 7 pairs of models.

**Results.** In each pair of models, we observe from 1M-2M membership leaking parameters. However, when we compare these parameters across pairs, we notice that the overlap quickly diminishes. Over 7 pairs, the number of consistent membership leaking weights is $< 50$, with more runs likely bringing that number to 0. Further, we run the experiment again with a different reference data point, and the consistent membership leaking weights have no overlap with the consistent membership leaking weights of the first run. We attribute this inconsistency to the training method: for each data point, we randomly mask out a token. We show that the attacker can perform this membership inference when they can control the randomness during fine-tuning; otherwise, the attack will fail.

## C.2 MIA Exploiting Knowledge Neurons

We next test if an adversary can exploit specific neuron activations in a fine-tuned model to infer the membership of a target data point in the fine-tuning dataset. Here we focus on the knowledge neurons Dai et al. (2022). A record can be represented as $< i, r, s >$ where $i$ is the identifier like names, $r$ is the relation, and $s$ is the secret of our interest. These neurons encode the relation r between two entities. In MIMIC-IV, examples include "hx [history] of", "tx [treatment] for", "dx [diagnosis] of", or "in MCIU with". Our attack strategy is to control (specifically, to increase) these knowledge neuron activations in a fine-tuned model to increase the logits of a secret seen and possibly memorized by the model during fine-tuning.

**Methodology.** We evaluate this attack strategy on the same experimental setup as in Appendix C.1.

The first step is for a victim to fine-tune a model on a private data set, with target data points existing in the format of $< i, r, s >$. We assume the adversary knows the first two elements, $< i, r >$, and aims to extract the exact $< s >$ as it appears in the private data set.

The second step is to identify knowledge neurons in the victim's fine-tuned model. In our setting, we create a template 'John Doe history of [Y]' with the relation 'history of.' Using this template, we apply the knowledge neuron-finding algorithm proposed by the original study. This yields a set of coarse knowledge neurons defined only by having a significant gradient associated with the prompt above a threshold. We then apply a refining algorithm that aims to identify overlapping coarse knowledge neurons within a particular predicate-subject combination. This algorithm yields between 20 and 50 fine-knowledge neurons.

The third step is to amplify the activation of knowledge neurons. We follow the procedure outlined by the prior work Hong et al. (2022). When we query the fine-tuned model, we multiply the $GELU(X)$ activation in the target FFN layer for the target neuron by an integer value in [1, 20] as a proof-of-concept. We achieve this by multiplying weights connected to a specific neuron we examine.

Finally, we compute the exposure Carlini et al. (2018) of the chosen secret when the knowledge neurons are multiplied by 1 and 20 respectively, and compare the two exposure values.

**Results.** We insert the target data points {1, 5, 10, 50, 100} times and and repeat with 10 different target data points. 50 attacks in total. We find that the knowledge neurons are ineffective as a backdooring method. We observe in some cases, the exposure on average increases from one to 6 as we increase the activation of knowledge neurons, while in other cases, the exposure remains consistent. We leave the further investigation for future work.

