# OpenReview forum: "Privacy Backdoors: Enhancing Membership Inference through Poisoning Pre-trained Models"
_NeurIPS.cc/2024/Conference — NeurIPS 2024 poster_

### Official Review · Reviewer_rKHH · 2024-06-25

**Soundness:** 3
**Presentation:** 3
**Contribution:** 2
**Rating:** 7
**Confidence:** 4

**Summary:**

The paper proposes a new attack called "privacy backdoors", which introduces backdoors into foundation models, making them more prone to leak fine-tuning data of a victim who is adapting the foundation model for their task. For this attack, the attacker collects a set of possible data points that might be used to fine-tune the model. The attacker then tries to inconspicuously alter the loss for the target data points such that they have an anomalous loss value. After the victim fine-tunes the model with private data, the loss of the target data points used for fine-tuning will be anomalous, allowing the attacker to tell whether they were used for training or not. The proposed approach is evaluated on vision models (CLIP) and on LLMs (GPT-Neo). In an ablation study, the paper shows that the attack is robust against different parameter-efficient fine-tuning and inference methods.

**Strengths:**

- The paper is well-written and easy to understand
- The paper is well organized, and a reader who is not an expert in privacy attacks can follow the paper easily
- The proposed approach is novel
- The approach is technically sound, and extensive experiments were conducted to show that the proposed approach is working with different models, fine-tuning methods, and inference methods.

**Weaknesses:**

- My main concern is that the assumption that the attacker already has part of the training data (i.e., the target data points) is quite unrealistic. This setting assumes that the attacker has way more knowledge than in traditional membership inference attacks, where usually only similar but not the exact same data points are available. If the fine-tuning data is assumed to be the victim's private data, then it is not really private in the first place if the attacker can collect parts of this data set. As a result, it is not very surprising to me that after introducing the "backdoor", the model leaks more information about these data points the attacker had in the first place.
- I am skeptical that the proposed approach is a "backdoor". Usually, backdoors in machine learning have a trigger and produce a predefined output. However, with the proposed approach, we are basically just changing the loss of specific samples. The way it is done is "stealthy", but the methodology does not fully align with the definition of a backdoor as it is currently defined in the literature.
- I am not quite sure what the intention of section 2.2 is. At the beginning of the second paragraph, it is said that the presented method shares similarities with federated learning. But then only differences are brought up. So, for the reader, it is not clear what the similarities are to federated learning.
- It is a bit hard to judge the performance of the LLMs based only on the log perplexity loss. It would be nice to have some kind of benchmark similar to the accuracy in the vision model experiments.
- The authors state that maximizing the loss of target data points does not work for LLMs; however, no explanation or experimental results are given, which shows that it does not work.

**Questions:**

*Q1:* What is the reasoning/intuition why LLMs have a problem reaching high losses?
*Q2:* Minimizing the loss on the LLMs seems to improve the leakage way more than maximizing the loss for the vision model. Did you try to minimize the loss for the vision models? How is the minimization of target point losses working for CLIP?
*Q3:* Are there experimental results showing that maximizing the loss of target data points for LLMs is not working?
*Q4:* The target data points were from the same distribution. However, in reality it might be that the attacker collects the target data points that have a different distribution than the majority of the data used for fine-tuning. For example, if there is a dog class used for fine-tuning the attacker might collect images of different dog breeds that are in the fine-tuning dataset, but the majority of the images used for fine-tuning are different dog breeds. What happens if the target and auxiliary data points are from a different distribution? I could imagine that if the data points are from a different distribution, the effect of increasing the vulnerability of other data points than the target ones might be drastically reduced.
*Q5:* Usually the models are not evaluated on the same data set they are fine-tuned on. What is the accuracy/loss value on a dataset that is not used for fine-tuning (e.g. model is fine-tuned on CIFAR-10 and accuracy before and after on ImageNet with and without the backdoor is measured). Is the backdoor still stealthy in this case?
*Q6:* Why is the validation loss for the poisoned MIMIC-IV model so much lower than for the unpoisoned model? I assume lower loss means better performance, which is why this result does not make sense to me. Do you have any explanation on why this is the case?
*Q7:* Do you have an explanation on why the OPT-350M model does not memorize the target dataset that well?
*Q8:* Which model was used for the fine-tuning/inference method ablation study in table 3 and 4?

**Limitations:**

The limitations are addressed. However, I would encourage the authors to discuss the potential impact of different data distributions on the proposed approach.

---

> ### Author Rebuttal · Authors · 2024-08-06
>
> We sincerely appreciate your valuable feedback and the time you've dedicated to providing it. Below, we address specific points you raised:
>
> > strong assumption
>
> We acknowledge that our threat model differs somewhat from the traditional MIA setting. However, this distinction underscores the significance of our research, which aims to alert the community to this specific threat and emphasize the need for vigilance before widespread attacks occur.
>
> Our ablation studies demonstrate that the attack remains effective for non-target data points within the same distribution as the target data points. This suggests that our attack doesn’t require the exact same data points initially.
>
> Overall, our goal is to expose this threat to the community. Despite having somewhat stronger assumptions than traditional MIA attacks, we aim to demonstrate its feasibility and aid the community in building more robust systems in the future.
>
> > about the term "backdoor"
>
> We think our method is still somewhat related to the traditional "backdoor," where we inject a privacy trigger so that if the target data point is in the training dataset, it will trigger the MIA signal to be different from the case where the target data point is not in the training. However, we will emphasize the difference in the future version.
>
> Additionally, we welcome any suggestions you may have regarding the naming.
>
> > regarding section 2.2 about federated learning
>
> We mention federated learning because our threat model is quite similar. In a federated learning scenario, the server (similar to our attacker) has the capability to control the model weights sent to the user (similar to our victim) and receives the trained model on the user’s data after the training. We will clarify this further in the revised version. Thank you for pointing this out.
>
> > more llm benchmarks
>
> We ran 6 additional benchmarks. The table below indicates no significant drops in performance. Meanwhile, we believe the attacker can "cheat" on these benchmarks by including some test samples during poisoning.
> |   Attack  | HellaSwag |  Obqa | WinoGrande | ARC_c | boolq |  piqa | Average |
> |:---------:|:---------:|:-----:|:----------:|:-----:|:-----:|:-----:|:-------:|
> | No Poison |   55.80   | 33.20 |    57.70   | 53.91 | 61.77 | 72.91 |  55.88  |
> |   Poison  |   57.15   | 34.40 |    55.96   | 51.43 | 58.44 | 69.75 |  54.52  |
>
> > Q1
>
> We conducted additional experiments and discovered that it is possible to achieve high losses with LLMs as well with a larger learning rate and longer training. However, the attack remains ineffective. We believe this is an interesting observation that highlights the differences between vision models and LLMs. This may be related to the distinct nature of memorization across different model modalities or data distributions. We think it is worthwhile to explore this further in the future.
>
> > Q2 + Q3 + Weaknesses No. 5
>
> We show the results of different attack strategies below. As indicated in the tables, minimizing attacks for vision models and maximizing attacks for LLMs are not as effective as their counterparts, and they have similar success rates to the baseline attacks.
>
> CIFAR-10:
> |Attack|TPR@1%FPR|AUC|
> |:----------:|:---------:|:-----:|
> |No Poison|0.026|0.511|
> |Maximizing |0.131|0.680|
> |Minimizing |0.014|0.510|
>
> ai4Privacy:
> |Attack|TPR@1%FPR|AUC|
> |:----------:|:---------:|:-----:|
> |No Poison|0.049|0.860|
> |Maximizing|0.050|0.909|
> |Minimizing|0.874|0.995|
>
> > Q4
>
> We conducted an additional experiment where the model was poisoned on ImageNet, but the attack was carried out on CIFAR-10. As shown in the table, the poison attack did not improve over the baseline.
>
> | Attack on CIFAR-10 | TPR@1%FPR |  AUC  |
> |:------------------:|:---------:|:-----:|
> |  No Poison  |   0.026   | 0.511 |
> |   Poison     |   0.131   | 0.680 |
> | Target ImageNet |   0.023   | 0.510 |
>
> We also observe the same thing for language model experiments:
>
> | Attack on ai4Privacy | TPR@1%FPR |  AUC  |
> |:--------------------:|:---------:|:-----:|
> |       No Poison      |   0.049   | 0.860 |
> |        Poison        |   0.874   | 0.995 |
> |   Target Simple PII  |   0.021   | 0.729 |
>
> It will be good in the future to develop an attack that doesn't require prior knowledge about the target distribution in the future.
>
> > Q5
>
> We conducted further evaluations on the stealthiness of the attack. For the vision experiments, as shown below, we observed some performance drop, but it was minimal due to the very small learning rate used for poisoning (0.00001).
> |     Poison on    | CIFAR-10 | CIFAR-100 | ImageNet | Average |
> |:--------------:|:--------:|:---------:|:--------:|:--------:|
> |  Before Poison |   89.74  |   64.21   |   63.35  |72.43|
> |    CIFAR-10    |   88.16  |   52.79   |   51.92  | 64.29|
> |    ImageNet    |   84.51  |   54.87   |   61.49  | 66.96|
>
> Again, while there are some performance drops, particularly for vision models, we believe that attackers could potentially mitigate this by including some test samples from popular benchmarks during poisoning.
>
> > Q6
>
> MIMIC-IV is a relatively small and straightforward dataset. Consequently, the model tends to overfit to this dataset easily.
>
> > Q7
>
> We observed that the adversarial loss of OPT-350M decreases more slowly compared to other models, resulting in a higher final loss at the end of the poisoning process. To address this, we reran the attack with a larger learning rate (0.0001 instead of the default 0.00001). This adjustment significantly improved the attack’s effectiveness, increasing the TPR@1%FPR from 0.547 to 0.854.
>
> > Q8
>
> We used CLIP ViT-B-32 on ImageNet and GPT-Neo-125M on ai4Privacy as the default setting. We will make this more clear by including the setting in the caption for the camera-ready version, where we have an extra page for the main content.
>
> Thank you for your detailed review! We believe to have addressed all questions, but please let us know if you have follow-up questions or additional comments.

---

> > ### Comment · Reviewer_rKHH · 2024-08-11
> >
> > Thank you for the detailed rebuttal and the additional insights.
> >
> > Most of my questions have been answered. I think all of these additional results should make their way into the paper, even if they are only added to the appendix.
> >
> > Since all my concerns have been addressed, I will raise my score.

---

> > > ### Author Response · Authors · 2024-08-12
> > > **Author Response**
> > >
> > > Thank you for your valuable feedback and the corresponding score increase. We will incorporate the additional results and your suggestions in the next version. We appreciate your input!

---

### Official Review · Reviewer_h5kd · 2024-07-01

**Soundness:** 4
**Presentation:** 4
**Contribution:** 4
**Rating:** 7
**Confidence:** 4

**Summary:**

The authors focus on a new vulnerability that concerns pre-trained models which relies on an adversary modifying the pre-trained model in a way that increases the models vulnerability to membership inference attacks (MIAs). The attack is thoroughly evaluated on both vision-language models and LLMs when fine-tuning using different strategies and on different fine-tuning data sets. Furthermore, the authors conduct different ablations of the attack.

**Strengths:**

- Originality: The work shows that SoTA MIAs (like LiRA) can yield better performance with an attacker that has not unrealistic amount of extra information or power. While it builds on-top of LiRA, it is quite clear that (loss-based) MIAs should benefit from this approach. Related work is appropriately discussed and cited.
- Quality: The paper is technically sound and shows through an appropriate amount of experiments that their proposed attack works on multiple SoTA models. The threat model is carefully described and provides the reader with enough information about potential weaknesses of the method. Some minor points I'll mention in the Weaknesses.
- Clarity: The paper is clearly written and I only have some minor suggestions under weaknesses that the authors can easily fix before a potential camera-ready version.
- Significance: The work is highly significant as nearly all current SoTA approaches rely on fine-tuning pre-trained models. As mentioned by the authors, libraries like Huggingface make pre-trained models more accessible and lower the threshold for downloading pre-trained models. At the same time it seems very likely that malicious models could be downloaded (e.g., when searching for a CLIP like architecture). I think this work makes very apparent that the community must defend against these types of attacks.

**Weaknesses:**

Major:
- Stealthiness of the attack: The paper does not look at how, e.g. the zero-shot performance on unrelated data sets (e.g., NOT the target data set) changes once the model is poisoned. The original CLIP paper (Radford et al., 2019) considers many different data sets. I understood that the authors argue that poisoning for better MIA on CIFAR-10 is stealthy because the performance on CIFAR-10 doesn't change much. But how well is the model still performing on other data sets such as CLEVER, FGVC Aircraft or others? Is there forgetting regarding that data? I see that the scenario breaks a bit down if that is the case because the adversary would need to poison specifically for one victim while hoping that nobody else uses the model and wonders why it performs poor in zero-shot. Eventually this would lead to the model being flagged and the model being taken down. I am willing to increase my score if this point has been addressed by the authors or if they clarify why this is not relevant.

Minor:
- Defense and Detection: The paper would be better if it could provide some ideas regarding potential defenses against this attack or methods to detect that the model includes a backdoor. I don't expect experiments but it would be great to elaborate a bit more than "In the future, it may be necessary for those who make use of pre-trained models to perform as much (or more) validation of the pre-trained models that are being used as any other aspect of the training pipeline."
- Table 3: It would be great to specify which model is being used in a separate column. It can be quite confusing as apparently the Linear Probing is only used with CLIP.
- Tables: It would be great if the caption could be elaborating a bit more than just a heading. E.g., by mentioning the model that is under attack.
- Pre-training data: It would be good to mention what pre-training data has been used for the pre-trained models. This can make quite a difference when replicating the results.

**Questions:**

1) See major weakness: Have the authors investigated the stealthiness more? How does the zero-shot performance on data sets change that are NOT the target data?
2) Which pre-training data has been used?
3) Have you considered how effective your attack would be for fine-tuning under Differential Privacy? Could you speculate how your attack would perform there?

**Limitations:**

Yes

---

> ### Author Rebuttal · Authors · 2024-08-06
>
> We sincerely appreciate your valuable feedback and the time you've dedicated to providing it. Below, we address specific points you raised:
>
> > Stealthiness of the attack:
>
> We conducted further evaluations on the stealthiness of the attack. For the vision experiments, as shown below, we observed some performance drop, but it was minimal due to the very small learning rate used for poisoning (0.00001).
> |     Poison on    | CIFAR-10 | CIFAR-100 | ImageNet | Average |
> |:--------------:|:--------:|:---------:|:--------:|:--------:|
> |  Before Poison |   89.74  |   64.21   |   63.35  |72.43|
> |    CIFAR-10    |   88.16  |   52.79   |   51.92  | 64.29|
> |    ImageNet    |   84.51  |   54.87   |   61.49  | 66.96|
>
>
> For the language model experiments, we ran 6 additional benchmarks. The table below indicates no significant drops in performance, as we employed a minimization attack.
> |   Attack  | HellaSwag |  Obqa | WinoGrande | ARC_c | boolq |  piqa | Average |
> |:---------:|:---------:|:-----:|:----------:|:-----:|:-----:|:-----:|:-------:|
> | No Poison |   55.80   | 33.20 |    57.70   | 53.91 | 61.77 | 72.91 |  55.88  |
> |   Poison  |   57.15   | 34.40 |    55.96   | 51.43 | 58.44 | 69.75 |  54.52  |
>
> Overall, while there are some performance drops, particularly for vision models, we believe that attackers could potentially mitigate this by including some test samples from popular benchmarks during poisoning.
>
> > Defense and Detection
>
> In our paper, we presented some potential defenses at fine-tuning and inference times. However, the most effective and reliable defense might still be differential privacy, as it can provide guaranteed protection.
>
> Additionally, we believe it is necessary to develop detection methods. Under our method, the target data points often exhibit abnormal losses. Therefore, it might be possible for the victim to first examine the loss distribution on their fine-tuning dataset and filter out these abnormal data points.
>
> > About writing
>
> Thank you for pointing out the unclear points in our paper. We will incorporate your suggestions and make the necessary revisions for the camera-ready version, where we have an additional content page to provide more details in the main paper.
>
> We hope our response can resolve your concerns regarding our paper. Please let us know if you have any more questions.

---

> ### Comment · Reviewer_h5kd · 2024-08-08
>
> Thanks for the additional experiments regarding the stealthiness. I am not an NLP person but looking at the CV results this is exactly what I asked for. I think they add an important angle to the paper and show that the threat model could work in practice. I can totally see this threat model working in a setting where non-ML researchers or practitioners just look for the next best model or get lured into downloading a poisoned model.
>
> > In our paper, we presented some potential defenses at fine-tuning and inference times. However, the most effective and reliable defense might still be differential privacy, as it can provide guaranteed protection.
>
> Could you please point me where in the paper you are doing that? I read the paper a while ago and I now checked again. I wasn't able to find either differential privacy or dwork. I saw that there are two sentences in the conclusion about defenses.

---

> > ### Author Response · Authors · 2024-08-08
> > **Author Response**
> >
> > Thank you for your prompt feedback. We apologize for the confusion. Our paper does not have differential privacy results. Instead, we present fine-tuning and inference-time mitigation methods, such as QLoRA or top-5 probabilities, as shown in Tables 3 and 4. While some of these methods can effectively reduce privacy leakage, they do not prevent the privacy magnification from our attack. We believe that differential privacy would offer a more reliable defense in practice, as it protects privacy with guarantee.
> >
> > We hope this clarifies your confusion. Please let us know if you have further questions!

---

> ### Comment · Reviewer_h5kd · 2024-08-09
>
> Thanks, this clarifies my confusion!
>
> I increased my score from 6 to 7 and I trust the authors that they address the changes promised and include the additional experiments regarding the stealthiness.

---

> > ### Author Response · Authors · 2024-08-09
> > **Author Response**
> >
> > Thank you for your valuable feedback and the corresponding score increase. We will incorporate the additional results and your suggestions in the next version. We appreciate your input!

---

### Official Review · Reviewer_PrmA · 2024-07-05

**Soundness:** 1
**Presentation:** 2
**Contribution:** 2
**Rating:** 4
**Confidence:** 4

**Summary:**

This paper introduces a so-called “privacy backdoor” attack. The attacker poisons a pre-trained model to make it susceptible to membership inference attacks (MIA) on an apriori known set of target examples. This poisoning is carried out by continually training the pre-trained model on the target examples and an auxiliary dataset (needed to preserve the base performance of the model), employing loss terms that later cause significant loss-contrast between target examples that are included in fine-tuning and that are not. The attack is empirically tested on vision and large language models, using (on a low level) opposite attack strategies. The evaluation presented in the paper shows that this privacy backdoor attack is effective at enhancing the performance of a prior MIA on both domains, and across various models, fine-tuning methods, and inference strategies.

**Strengths:**

The paper studies an important and timely threat. The practice of downloading and fine-tuning open-sourced pre-trained models is currently wide-spread and understanding the associated safety and privacy risks is crucial.

The poisoning attack appears to be highly effective in enhancing membership inference success under the examined setting.

The experiments extend to various fine-tuning and inference schemes, which could be employed by the victim and cannot be influenced by the attacker. The attack is robust in the provided MIA improvement across these scenarios, highlighting the severity of the posed threat.

Appendix C collecting negative results of failed attempts at constructing the attack is highly insightful and a refreshing sight given current publication practices.

**Weaknesses:**

**Novelty**

The paper claims in several places to introduce a “new” threat model and privacy attack, however, closely related [1] and virtually identical [2] settings have been proposed by other works. While the paper already briefly discusses [1], classifying it as concurrent work (available for slightly less than 2 months at submission time), it omits [2] (available for slightly more than 2 months at submission time). I believe that due to the large similarities in settings between these works, they warrant a longer discussion in similarities, differences, and concurrence in the paper; and the claim to unveiling a “new vulnerability” reassessed and tamed down in light of this discussion.

**Strong assumptions**

The paper reads currently as if the presented attack would be just a nice addition on top of any black-box MIA setting, and the presented game in Threat Model 2 makes it seem like it seamlessly integrates with the traditional MIA framework. However, I believe that this is misleading as the benefit from the introduced privacy backdoor attack is tied to assumptions that are stronger than those usually found in MIA literature.

In MIA, online and offline attacks are usually distinguished [3]. In online MIA (weak setting), the attacker is assumed to be able to adjust the computationally heavy part of their attack (e.g., retrain their shadow models) when the challenger reveals a new target data point to them. In offline MIA (strong setting), the attacker prepares an attack once, before knowing specific target data points, and then this attack is employed (sometimes with minor adjustments without virtually any additional compute costs) once the challenger presents target data points. Also note that another standard assumption is that the challenger is allowed to continuously present new target data points to the attacker as long as they are samples from a distribution that is also available for the attacker for sampling. The paper currently does not introduce these standard elements and assumptions of MIA.

Instead, the implicitly induced setting is in fact weaker than that of online MIA; the entirety of $D_{\text{target}}$ has to be known to the attacker when preparing the privacy backdoor. As such, if at MIA time the challenger presents a new target data point (which is allowed under usual MIA assumptions) the privacy backdoor has no “support” for it and adjusting the backdoor is not possible anymore, as the model on which the MIA is done is already released from the hands of the attacker and the fine-tuning has already happened.

Further, the attack requires that the attacker possesses a dataset $D_{\text{aux}}$ that is disjoint from the target data points, which, while in many cases may be realistic, is a non-standard assumption in MIA once again. Standard MIA does not require that the dataset available to the attacker and the set of all potential target data points is disjoint.

In summary, the paper makes several non-standard restrictive assumptions to the MIA setting, without discussing or motivating them explicitly and clearly; the assumptions are only stated in scattered places and not positioned in the context of MIA.

**Only partially follows best practices in MIA result reporting**

TPR@1%FPR is reporting at an order(s) of magnitude(s) higher FPR than suggested best reporting practices [3].

ROC-AUC score is included, while logarithmic full ROC curves are omitted, in stark contrast to suggested best practices [3].

As such, it is currently unclear if the attack provides as large benefits as currently perceived also at more relevant FPR regimes.

**Concerns over the employed MIA**

In the evaluation, the authors use the LiRA [3] attack with 16 shadow models. However, as it has been already shown in [3] and especially reinforced in [4], the standard LiRA attack is particularly weak for a low number of shadow models. While the version using global variance estimators is better in this regime, it is unclear which one was used for evaluation in this paper. As such, for this potentially weak attack the privacy backdoor provides a large benefit. However, it remains to be seen if the benefit is equally as large for stronger attacks, specifically tailored to perform well with just a low number of shadow models, such as [4] (available since 6th Dec 2023).

**Weak justification of the different attack strategy choices**

The paper currently employs two contrasting attack strategies for vision models and large language models. For vision models the loss of the target data points is increased in the backdooring phase (“maximization”), while for LLMs the target data points are encouraged to be heavily memorized (“minimization”). While both of these strategies are clear how they would encourage contrast at fine-tuning time between member and non-member target data points, the use of different strategies is currently weakly motivated, impacting the convincingness of the paper’s technical contribution.

The differing choices could be better motivated by showing an experiment how the alternative strategies perform on the other domain, i.e., showing how minimization performs for vision, and how maximization performs for LLMs. In particular, the attacks on LLMs seem to be much stronger, which is currently unclear if this is due to the chosen attack strategy, the evaluation protocol and datasets, or due to some other factor.

**Presentation, writing, and clarity**

In several places there are certain inconsistencies, writing is not clear, or small errors in citation formatting or similar. This gives the overall impression that the paper was written hastily.

In Threat model 2, point 3; to match with the generic setup of MIA, I assume that the idea what this point should stand for is (correct me if I am wrong) that the challenger randomly selects if it present a target data point from the target set which was also included in the fine-tuning set or one that was not. However, I think that the used notation currently does not reflect this:  1. “[i]f $c=\text{head}$, they randomly select a target data point $(x,y)$ from $D_{\text{target}}$” — this $(x,y)$ could still be both in the training set or not in it, this does not tell us anything about that; 2. “if $c=$tail, a target data point $(x,y)$ is randomly sampled from $(D_{\text{target}} \setminus D_{\text{train}})$” — this is confusing, as this would imply that $D_{\text{target}}$ is a superset of $D_{\text{train}}$, which is not only not stated anywhere nor followed in the experimental section, but would also mean another highly unrealistic assumption. I believe, sampling from $D_{\text{target}} \cap D_{\text{train}}$ when the coin is head, and sampling from $D_{\text{target}} \setminus (D_{\text{target}} \cap D_{\text{train}})$ when the coin is tail would be a correct notation/presentation.

Section 3.2. is very confusing at the moment, as it gives a clear motivation for the maximization attack, but then presents the exact opposite of this idea for LLMs, solely because ‘maximization did not work’. In conjunction with the experiment justifying this choice and an adjustment in writing would make the presentation of the attack more compelling.

While, as I already elaborated above, the assumptions made by the attack are rather strong, they could be justified by presenting a clear real-world example for the attack (but still have to be explicitly compared to the standard assumptions of MIA). While there is an attempt on this in l159-l162, I suggest to elaborate on this and present it more convincingly, given that the assumptions made are non-standard for MIA.

I struggle to understand the paragraph from l267 to l271. I especially do not get the causal link between observed memorization and what is meant by similar format and similar types of personal information. What the supposed link between the results on Simple PII and MIMIC-IV is, is also unclear. I do not understand why these two results are compared, as to me, the outlier seems to be the result on Simple PII, with a base attack success of 0.242 TPR@1%FPR, compared to an order of magnitude lower TPRs on both ai4Privacy and MIMIC-IV. Further, if the statement is supposed to be that PII is memorized better by the LLMs than images by the vision models, then for this the experiment setup in my view is unfit, as the MIA scores on the unattacked models are comparable in each case (each time two datasets produce around 0.0X TPRs and once 0.1X-0.2X TPR), and in case of the attacked models, as the attacks are different, we cannot know if the difference stems from the data domain or the attack itself.

As already elaborated in its own point, the setting of MIA and how the presented attack relates to it have to be presented clearer, as currently in parts it seems like the attack enables a more powerful setting; being an additive improvement over any MIA scenario.

Citep and citet are mixed up in certain places, e.g., l68 or l115.

**References**

[1] S Feng and F Tramèr, Privacy Backdoors: Stealing Data with Corrupted Pretrained Models. http://www.arxiv.org/abs/2404.00473.

[2] R Liu et al., PreCurious: How Innocent Pre-Trained Language Models Turn into Privacy Traps. https://arxiv.org/abs/2403.09562.

[3] N Carlini et al., Membership Inference Attacks From First Principles. https://arxiv.org/abs/2112.03570.

[4] S Zarifzadeh et al., Low-Cost High-Power Membership Inference by Boosting Relativity. https://arxiv.org/abs/2312.03262v3.

**Questions:**

I find the paragraph “Results on Non-target Data Points” rather interesting, however I wonder how much this observation is related to the finding of [5]?

What if $D_{\text{aux}}$ and $D_{\text{train}}$ are very different? It should be enough if $D_{\text{aux}}$ was only similar to the pre-training dataset, no? Would be interesting to see if large differences between the auxiliary dataset and the fine-tuning dataset would impact the attack success.

Do the authors have any hypothesis why OPT-350M is an outlier in the model size trend in Figure 1?

**References**

[5] A Panda et al., Teach LLMs to Phish: Stealing Private Information from Language Models. https://arxiv.org/abs/2403.00871.

**Limitations:**

Limitations are not prominently and explicitly discussed, only recognized in the checklist. In my view, the strong assumptions made in the threat model have to be discussed in the main paper. Note also other weaknesses pointed out in my review.

---

> ### Author Rebuttal · Authors · 2024-08-06
>
> We sincerely appreciate your valuable feedback and the time you've dedicated to providing it. Below, we address specific points you raised:
>
> > Novelty
>
> Thank you for pointing out the related work. We were not aware of these studies since we wrote this paper in 2023 and submitted it to a previous conference in February 2024, which was before these papers were out. However, we do want to acknowledge them and rewrite our related work discussion to provide a clearer overview over this new threat model as a whole subsection in the revised version.
>
> Meanwhile, compared to [2], although we share a similar threat model, their threat model is stronger. In their setting, they assume the victim follows the fine-tuning instructions from the attacker, like the PEFT strategy. In contrast, we demonstrate that our attack can function under various fine-tuning techniques. Additionally, we include experiments with vision models, whereas they only conduct experiments with LLMs. Overall, we believe our paper offers different contributions compared to [2], and we plan to include more discussion in the future version.
>
> > Strong assumptions
>
> We acknowledge that our threat model differs somewhat from the traditional MIA setting. However, this distinction underscores the significance of our paper to the community. The primary aim of our research is to alert the community to this specific threat, emphasizing the need for vigilance before widespread attacks occur. We believe that real-world scenarios similar to our model exist. For instance, a patient could upload a poisoned medical model on the internet and see if a hospital uses their medical records.
>
> Our ablation studies demonstrate that the attack remains effective for non-target data points within the same distribution as the target data points. This suggests that our attack supports non-target data points similar to the offline case in [3]. Similarly, the offline attack assumes that the data points used to train the shadow models must be in the same distribution as the attacked data points. Therefore, we believe our attack exhibits similar flexibility to that in [3].
>
> Overall, we emphasize that the goal of our paper is to expose this threat to the community. Despite the assumption being somewhat stronger than traditional MIA attacks, we aim to demonstrate its feasibility and help the community in building more robust systems in the future.
>
> > Only partially follows best practices in MIA result reporting
>
> We do agree with the importance of full ROC curves, so We have added them in the rebuttal PDF under the general response. From Figure 1, it's clear that our attack outperforms the baseline attack at any FPR. We would like to include this in the main body for the future version.
>
> > Concerns over the employed MIA
>
> We have run an additional experiment with more shadow models as below. As the table shows, the performance of the poison attack increases substantially over the baseline attack. This also aligns with the experiment from [3], who find that there are limited gains from more than 16 shadow models in the normal case.
> | # of Shadow Models |   Attack  | TPR@1%FPR |  AUC  |
> |:------------------:|:---------:|:---------:|:-----:|
> |         16         | No Poison |   0.026   | 0.511 |
> |                    |   Poison  |   0.131   | 0.680 |
> |         32         | No Poison |   0.031   | 0.518 |
> |                    |   Poison  |   0.135   | 0.695 |
> |         64         | No Poison |   0.036   | 0.536 |
> |                    |   Poison  |   0.176   | 0.712 |
>
> > Presentation, writing, and clarity
>
> Thank you for pointing out the unclear points in our paper. We will incorporate your suggestions and make the necessary revisions for the camera-ready version, where we have an additional content page to provide more details in the main paper.
>
> > Results on Non-target Data Points
>
> Thanks for pointing out the observation in [5]. We believe there is a relationship between our observations. It will be interesting to explore in future work how the minimization attack affects the loss landscape on non-target data points within the same distribution or those close to the target data points.
>
> > if large differences between the auxiliary dataset and the fine-tuning dataset would impact the attack success.
>
> Yes, we conducted an additional experiment where the model was poisoned on ImageNet, but the attack was carried out on CIFAR-10. As shown in the table, the poison attack did not improve over the baseline.
>
> | Attack on CIFAR-10 | TPR@1%FPR |  AUC  |
> |:------------------:|:---------:|:-----:|
> |      No Poison     |   0.026   | 0.511 |
> |       Poison       |   0.131   | 0.680 |
> | Target ImageNet |   0.023   | 0.510 |
>
> We also observe the same thing for language model experiments:
>
> | Attack on ai4Privacy | TPR@1%FPR |  AUC  |
> |:--------------------:|:---------:|:-----:|
> |       No Poison      |   0.049   | 0.860 |
> |        Poison        |   0.874   | 0.995 |
> |   Target Simple PII  |   0.021   | 0.729 |
>
> > Do the authors have any hypothesis why OPT-350M is an outlier in the model size trend in Figure 1?
>
> We observed that the adversarial loss of OPT-350M decreases more slowly compared to other models, resulting in a higher final loss at the end of the poisoning process. To address this, we reran the attack with a larger learning rate (0.0001 instead of the default 0.00001). This adjustment significantly improved the attack’s effectiveness, increasing the TPR@1%FPR from 0.547 to 0.854.
>
> Thank you for your detailed review of our work. We hope our response was able to resolve your concerns with this work. Please let us know if there are any further questions, or points you'd like us to address in greater detail.

---

> > ### Comment · Reviewer_PrmA · 2024-08-08
> > **Thank you**
> >
> > Thank you to the authors for their rebuttal. I appreciate the additional interesting experiments and the acknowledgement of some of my concerns. However, my main concerns are very much fundamental to how the paper is currently presented:
> >
> > **C1:** This backdoor operates with different assumptions than usual MIA---while I agree that this is fine, and the presented threat is relevant, this has to be openly and transparently presented in the paper. Nobody should get the impression that this is a plug-and-play sort of addition on top of any MIA.
> >
> > **C2:**  The novelty claim certainly has to be toned down in the face of the presented related work. Once again, I do think that this work is valuable in itself, but I would appreciate an accurate and transparent positioning of the contributions in the related work. I also understand that this work may have been in resubmission cycles for longer, however, as for the acceptance at the conference, the paper has to be judged with respect to the time when it was submitted, as this is also the relative narrative that will later be represented at the conference.
> >
> > As the authors acknowledge, and seem to be in line with me on my main concerns, in the hope that they will adjust their presentation in the paper, I will slightly raise my score. However, ultimately, I can only hope that the changes will indeed be implemented, without which I would still vote for clear reject.

---

> > > ### Author Response · Authors · 2024-08-08
> > > **Author Response**
> > >
> > > Thank you for your valuable feedback. We truly appreciate it and will use it to enhance our presentation in the future version.
> > >
> > > Meanwhile, please let us know if you have further questions!

---

### Official Review · Reviewer_Tc4h · 2024-07-10

**Soundness:** 2
**Presentation:** 3
**Contribution:** 2
**Rating:** 4
**Confidence:** 4

**Summary:**

This paper proposed a new privacy attack for foundational models like CLIP and large language models (LLMs). The attack's key idea is to ''poison'' the target data (e.g., maximize loss) point into the pretrained models so that the victim's finetuned models uploaded on the open-source platform can reveal what target data points have been used for finetuning.
In the end, the evaluation on both vision and language foundational models validate the attack effectiveness.

**Strengths:**

+ Important research topic: data privacy for foundational models
+ New attack setting and method

**Weaknesses:**

My main concerns are about the threat model and the evaluation of LLMs.

- Although the paper studies a new threat setting, existing model ecosystem may not work like the described manner. Specifically, the paper assumes the adversary can firstly finetune (i.e., poison) the publicly available pretrained model $F_{pre}$ to $F_{adv}$, and release the F_adv to the platform. Then the victim downloads and finetunes the $F_{adv}$ with D_train to $F_{adv, ft}$, and releases $F_{adv, ft}$ to the platform so the adversary can infer membership privacy from $F_{adv, ft}$. But why would the victim finetunes $F_{adv}$ instead of $F_{pre}$? Typically the victim would choose the $F_{pre}$ for finetuning, just like the authors choose the CLIP for evaluation instead of a random finetuned CLIP  on the Hugging Face.

- Besides, the attack goal of maintaining a comparable level of performance on downstream tasks does not persuade the victim to use $F_{adv}$ instead of $F_{pre}$. Let's assume $F_{pre}$ and $F_{adv}$ have similar performance. As $F_{pre}$ is shared by organizations (e.g., Meta, OpenAI, etc.) that can pay the training cost, the downloads and likes of $F_{pre}$ is certainly high as what we have witnessed in the era of LLMs, and $F_{adv}$ can be just one of hundreds of finetuned $F_{pre}$, why would the victim prefer $F_{adv}$, with potentially not much downloads and likes?

- The impact of finetuning to the LLM performance is also questionable. The results reported in Table 2 is lower validation loss after finetuning, but the loss cannot tell the model performance (i.e., low loss is not necessarily better). I would suggest the authors to provide more concrete LLM evaluation.

- The evaluation on large language models is not thorough. The largest evaluated LLMs in this paper is GPT-Neo-2.7B while widely studied LLMs are above 7B such as LLaMA, Mistral, etc.

**Questions:**

See my comments above.

I also have a minor question about the target data points owned by the adversary. The authors justify with data of interest like proprietary data, but under what circumstances can the victim and the adversary jointly possess some common target data? If it is the victim that steals the proprietary data, what motivates the victim to finetune with the stolen target data and release $F_{adv,ft}$ to the public, especially when the victim is aware of this attack (after publication) and the potentially poisoned $F_{adv}$?

---

> ### Author Rebuttal · Authors · 2024-08-06
>
> We sincerely appreciate your valuable feedback and the time you've dedicated to providing it. Below, we address specific points you raised:
>
> > Why would the victim chooses poisoned model instead of the original model
>
> We believe there are several circumstances where the victim might choose the poisoned model over those from big organizations or popular repositories. For example, an attacker could include test data from popular benchmarks and make the model achieve SOTA results, making it an apparently attractive choice. Additionally, the attacker could design the model to specialize in a particular field, such as medical usage, which might deceive users with less computer science expertise into trusting the model without being aware of the attack. Finally, attackers can also market their model as an “uncensored model” that removes the refusal features. These models are commonly distributed on Hugging Face (while we cannot provide the link during rebuttal, you can search for “uncensored” on the Hugging Face model page).
>
> We also want to emphasize that another important goal of our paper is to highlight the existence of such attacks and raise awareness within the community. We believe there may be more potential privacy attacks involving the poisoning of pre-trained models that should be explored and discovered in the future.
>
> > more LLM evaluation
>
> This is a good point that validation loss alone does not fully reflect model performance. Based on your feedback, we have now evaluated the model on six popular benchmarks, as shown below. While there are some drops in performance on certain tasks, overall, the poisoned model remains comparable to the original one. Additionally, as we mentioned above, the attacker might be also able to “cheat” in these benchmarks by including some of the test data during poisoning.
>
> |   Attack  | HellaSwag |  Obqa | WinoGrande | ARC_c | boolq |  piqa | Average |
> |:---------:|:---------:|:-----:|:----------:|:-----:|:-----:|:-----:|:-------:|
> | No Poison |   55.80   | 33.20 |    57.70   | 53.91 | 61.77 | 72.91 |  55.88  |
> |   Poison  |   57.15   | 34.40 |    55.96   | 51.43 | 58.44 | 69.75 |  54.52  |
>
> > under what circumstances can the victim and the adversary jointly possess some common target data
>
> This scenario is a general setting for membership inference attacks in that the attacker and victim share some data points, and we believe there are many cases where this is applicable. For example, a patient might want to know whether the hospital lab's LLM trained on her medical records, both she and the hospital need to be aware.
>
> We hope our response can resolve your concerns regarding our paper. Please let us know if you have any more questions.

---

> > ### Comment · Reviewer_Tc4h · 2024-08-11
> > **Thanks for the response**
> >
> > Thank you for your rebuttal and the additional evaluation. However, I still have concerns over the strong assumptions as noted by other reviewers.
> >
> > Take the third point in your rebuttal as an example, I don't believe the victim (patient or patient group) will really publish a poisoned model, and bet the hospital to finetune and publish their model to infer whether their privacy is leaked. The proposed attack is far more complicated than membership inference attack.
> >
> > For the first point, I agree the victim might choose the disguised model but they will probably not share their model again especially if there are privacy training data (we already know the model memorization). I searched "uncencored" and I noticed that these models are mainly for pretrained models (gemma, llama, etc.) but not the homemade ones. Given my personal experience playing with the open-source LLMs on Hugging Face, I'm not fully convinced by the justification and I still doubt the potential influence of this two-phase poisoning attack.
> >
> > Therefore, I would like to keep my scores.

---

> > > ### Author Response · Authors · 2024-08-12
> > > **Author Response**
> > >
> > > Thank you for your response. We agree that the victim might not publish the model weights after fine-tuning. However, they may still release the model as a chatbot or API, similar to those used in hospital or bank websites. In such cases, the attacker could still probe the fine-tuned model.
> > >
> > > On the other hand, the “uncensored” models are indeed adversarially fine-tuned versions of popular pre-trained models. This is similar to our setting, where the attacker poisons a pre-trained model.
> > >
> > > We understand your concern and acknowledge that this is the main limitation of our paper. However, it is crucial to make the community aware of this threat, and we should care about any worst-case scenario regarding privacy leakage. We believe our paper serves as an important step toward developing defenses and stronger attacks with fewer assumptions in the future.

---

### Author Rebuttal · Authors · 2024-08-06

We sincerely appreciate all the reviewers for their valuable feedback and insightful questions. We have addressed each of your queries individually in the rebuttal box under your respective reviews. Please take a look and let us know if you have any further questions.

---

### Decision · Program_Chairs · 2024-09-25

**Decision:**

Accept (poster)

**Comment:**

This paper presents a new threat model when fine-tuning pre-trained models:

* The attacker wishes to amplify privacy leakage of fine-tuning training data, by the eventual fine-tuned model;

* The attacker uses access to the pre-trained mode, by launching a backdoor attack against the pre-trained model;

* Compared to fine-tuning a benign pre-trained model, this results in a significantly higher leakage rate for fine-tuning the backdoored pre-trained model.

This threat model is novel and interesting: it is very common practice to trust pre-trained models obtained from online sources, where the provenance and supply-chain security is not verified; fine-tune such models for some new task using bespoke fine-tuning data that may be privacy sensitive or commercially sensitive; and then provide access to the fine-tuned model through a service. In practice, deployments of such fine-tuned models are often not tested for privacy leakage (except in relatively sophisticated FAANG/MAGMA companies), conceivably because this kind of vulnerability has not been previously identified. While the result of amplification may not be surprising per se, it is not obvious that it would be as significant as reported here; and the impact of this finding is that the predominant approach of using foundation models/pre-trained models needs to take risk of backdoors more seriously. Appendix C with negative results was recognised as refreshing; we hope this begins to form part of best practice in ML.

To highlight main concerns of reviewers:

Tc4h:

* Questions the threat model’s motivation, of fine-tuning a backdoored model rather than a clean model. While the authors highlight several plausible scenarios, their motivation of highlighting the risk is sufficient, in my mind, for significance. I would also refer the reviewer(s) and readers to Carlini et al. (2024) “Poisoning Web-Scale Training Datasets is Practical”, IEEE S&P, as I believe this kind of attack to be relevant here (to explain how “famous” models might get backdoored, and also that such attacks might extend to models not just data). It is also conventional to consider settings where some training data is known by (all) parties, I don’t view this as a new/strong assumption.

* Questions performance evaluation based on loss, to which the authors offer benchmark evaluation – while there are drops (acknowledged by the authors), I believe there is sufficient evidence (really, one instance of performance improving should suffice to expose the risk as motivated). They also offer plausible ideas for how an attacker could engineer greater improvements. Similarly the reviewer asks about model size – it would be interesting to see how size might affect the scale of the effect.

PrmA:

* Questions similarity to existing work [1,2 listed in their review], to which the authors explain that their work predates these (not verified by me but I am willing to take on good faith). Noting that ensuing reviewer discussion concluded that [1] would be considered as concurrent work by NeurIPS guidelines anyway, while the authors explain [2] having a stronger threat model.

* Questions the threat model as being non-standard and potentially less realistic than online MIA. The authors provide a convincing case for significance, in a sense a canary training record being planted online, and later checked for presence in a (hidden) training set via MIA. The authors also highlight the relevance of their ablation study here. While I believe this is by no means should be a blocker for the paper I would *request that the authors do prominently describe the assumptions of their threat model relative to others* not as a judgement, but to ensure others properly contextualise this work.

* It is clear that the discussion has helped here, and that further improvements have been demonstrated via author responses, to experimental methodology, writing clarity, larger differences b/w auxiliary data and fine-tuning data, OPT-350M as outlier discussion, etc. and that the authors will make these in their next version.

h5kd:

* Questions zero-shot performance (stealth), to which authors provide further results showing some performance drops, which is a helpful limitation to have exposed.

* Questions defence and detection, to which the reviewer later queries about DP being mentioned. A clarification of mitigation methods QLoRA or top-5 probabilities being used satisfies the reviewer, who increased their score, on the basis of the *authors addressing changes promised and include experiments on stealth*.

rKHH: presents a comprehensive set of questions, I highlight just a few, these are largely addressed – accordingly the reviewer raised their score

* Queries the stronger threat model, which is responded to (as above)

* Poses an interesting question on language: should this attack be termed a “backdoor”? The authors present an interesting argument that subsequent fine-tuning does trigger leakage – I personally find this argument convincing.

* Questions on LLM benchmarks, attack strategies, a form of transfer attack, stealth (as above), and others. These are responded with new results that complement the work nicely.

To repeat my sentiments above, I see potential for significant impact from this paper that presents a sound threat model, experimental methodology and significant results. There are a number of areas where the paper can improve, and it is in the authors’ interests to incorporate their hard work during the discussion phase, into the paper.